# Scalable Demand-Aware Recommendation

Jinfeng Yi[1]*, Cho-Jui Hsieh[2], Kush R. Varshney[1], Lijun Zhang[3], Yao Li[2]

[1]IBM Thomas J. Watson Research Center, Yorktown Heights, NY, USA
[2]University of California, Davis, CA, USA
[3]National Key Laboratory for Novel Software Technology, Nanjing University, Nanjing, China
jinfengyi.ustc@gmail.com, chohsieh@ucdavis.edu, krvarshn@us.ibm.com,
zhanglj@lamda.nju.edu.cn, yaoli@ucdavis.edu

## Abstract

Recommendation for e-commerce with a mix of durable and nondurable goods has characteristics that distinguish it from the well-studied media recommendation problem. The demand for items is a combined effect of *form utility* and *time utility*, i.e., a product must both be intrinsically appealing to a consumer and the time must be right for purchase. In particular for durable goods, time utility is a function of inter-purchase duration within product category because consumers are unlikely to purchase two items in the same category in close temporal succession. Moreover, purchase data, in contrast to rating data, is implicit with non-purchases not necessarily indicating dislike. Together, these issues give rise to the positive-unlabeled demand-aware recommendation problem that we pose via joint low-rank tensor completion and product category inter-purchase duration vector estimation. We further relax this problem and propose a highly scalable alternating minimization approach with which we can solve problems with millions of users and millions of items in a single thread. We also show superior prediction accuracies on multiple real-world datasets.

## 1 Introduction

E-commerce recommender systems aim to present items with high utility to the consumers [18]. Utility may be decomposed into *form utility*: the item is desired as it is manifested, and *time utility*: the item is desired at the given point in time [28]; recommender systems should take both types of utility into account. Economists define items to be either *durable goods* or *nondurable goods* based on how long they are intended to last before being replaced [27]. A key characteristic of durable goods is the long duration of time between successive purchases within item categories whereas this duration for nondurable goods is much shorter, or even negligible. Thus, durable and nondurable goods have differing time utility characteristics which lead to differing demand characteristics.

Although we have witnessed great success of collaborative filtering in media recommendation, we should be careful when expanding its application to general e-commerce recommendation involving both durable and nondurable goods due to the following reasons:

1. Since media such as movies and music are nondurable goods, most users are quite receptive to buying or renting them in rapid succession. However, users only purchase durable goods when the time is right. For instance, most users will not buy televisions the day after they have already bought one. Therefore, recommending an item for which a user has no immediate demand can hurt user experience and waste an opportunity to drive sales.

2. A key assumption made by matrix factorization- and completion-based collaborative filtering algorithms is that the underlying rating matrix is of low-rank since only a few factors typically contribute to an individual's form utility [5]. However, a user's demand is not only driven by form utility, but is the combined effect of both form utility and time utility. Hence, even if the underlying form utility matrix is of low-rank, the overall purchase intention matrix is likely to be of high-rank,[2] and thus cannot be directly recovered by existing approaches.

An additional challenge faced by many real-world recommender systems is the *one-sided sampling* of implicit feedback [15, 23]. Unlike the Netflix-like setting that provides both positive and negative feedback (high and low ratings), no negative feedback is available in many e-commerce systems. For example, a user might not purchase an item because she does not derive utility from it, or just because she was simply unaware of it or plans to buy it in the future. In this sense, the labeled training data only draws from the positive class, and the unlabeled data is a mixture of positive and negative samples, a problem usually referred to as *positive-unlabeled (PU) learning* [13]. To address these issues, we study the problem of *demand-aware recommendation*. Given purchase triplets (user, item, time) and item categories, the objective is to make recommendations based on users' overall predicted combination of form utility and time utility.

We denote purchases by the sparse binary tensor $\mathcal{P}$. To model implicit feedback, we assume that $\mathcal{P}$ is obtained by thresholding an underlying real-valued utility tensor to a binary tensor $\mathcal{Y}$ and then revealing a subset of $\mathcal{Y}$'s positive entries. The key to demand-aware recommendation is defining an appropriate utility measure for all (user, item, time) triplets. To this end, we quantify purchase intention as a combined effect of form utility and time utility. Specifically, we model a user's time utility for an item by comparing the time $t$ since her most recent purchase within the item's category and the item category's underlying inter-purchase duration $d$; the larger the value of $d - t$, the less likely she needs this item. In contrast, $d \leq t$ may indicate that the item needs to be replaced, and she may be open to related recommendations. Therefore, the function $h = \max(0, \ d - t)$ may be employed to measure the time utility factor for a (user, item) pair. Then the purchase intention for a (user, item, time) triplet is given by $x - h$, where $x$ denotes the user's form utility. This observation allows us to cast demand-aware recommendation as the problem of learning users' form utility tensor $\mathcal{X}$ and items' inter-purchase durations vector $\mathbf{d}$ given the binary tensor $\mathcal{P}$.

Although the learning problem can be naturally formulated as a tensor nuclear norm minimization problem, the high computational cost significantly limits its application to large-scale recommendation problems. To address this limitation, we first relax the problem to a matrix optimization problem with a label-dependent loss. We note that the problem after relaxation is still non-trivial to solve since it is a highly non-smooth problem with nested hinge losses. More severely, the optimization problem involves $mnl$ entries, where $m$, $n$, and $l$ are the number of users, items, and time slots, respectively. Thus a naive optimization algorithm will take at least $O(mnl)$ time, and is intractable for large-scale recommendation problems. To overcome this limitation, we develop an efficient alternating minimization algorithm and show that its time complexity is only approximately proportional to the number of nonzero elements in the purchase records tensor $\mathcal{P}$. Since $\mathcal{P}$ is usually very sparse, our algorithm is extremely efficient and can solve problems with millions of users and items.

Compared to existing recommender systems, our work has the following contributions and advantages: (i) to the best of our knowledge, this is the first work that makes demand-aware recommendation by considering inter-purchase durations for durable and nondurable goods; (ii) the proposed algorithm is able to simultaneously infer items' inter-purchase durations and users' real-time purchase intentions, which can help e-retailers make more informed decisions on inventory planning and marketing strategy; (iii) by effectively exploiting sparsity, the proposed algorithm is extremely efficient and able to handle large-scale recommendation problems.

## 2  Related Work

Our contributions herein relate to three different areas of prior work: consumer modeling from a microeconomics and marketing perspective [6], time-aware recommender systems [4, 29, 8, 19], and PU learning [20, 9, 13, 14, 23, 2]. The extensive consumer modeling literature is concerned with descriptive and analytical models of choice rather than prediction or recommendation, but nonetheless

forms the basis for our modeling approach. A variety of time-aware recommender systems have been proposed to exploit time information, but none of them explicitly consider the notion of time utility derived from inter-purchase durations in item categories. Much of the PU learning literature is focused on the binary classification problem, e.g. [20, 9], whereas we are in the collaborative filtering setting. For the papers that do examine collaborative filtering with PU learning or learning with implicit feedback [14, 23, 2, 32], they mainly focus on media recommendation and overlook users' demands, thus are not suitable for durable goods recommendation.

Temporal aspects of the recommendation problem have been examined in a few ways: as part of the cold-start problem [3], to capture dynamics in interests or ratings over time [17], and as part of the context in context-aware recommenders [1]. However, the problem we address in this paper is different from all of those aspects, and in fact could be combined with the other aspects in future solutions. To the best of our knowledge, there is no existing work that tries to take inter-purchase durations into account to better time recommendations as we do herein.

# 3 Positive-Unlabeled Demand-Aware Recommendation

Throughout the paper, we use boldface Euler script letters, boldface capital letters, and boldface lower-case letters to denote tensors (e.g., $\mathcal{A}$), matrices (e.g., $\mathbf{A}$) and vectors (e.g., $\mathbf{a}$), respectively. Scalars such as entries of tensors, matrices, and vectors are denoted by lowercase letters, e.g., $a$. In particular, the $(i, j, k)$ entry of a third-order tensor $\mathcal{A}$ is denoted by $a_{ijk}$.

Given a set of $m$ users, $n$ items, and $l$ time slots, we construct a third-order binary tensor $\mathcal{P} \in \{0, 1\}^{m \times n \times l}$ to represent the purchase history. Specifically, entry $p_{ijk} = 1$ indicates that user $i$ has purchased item $j$ in time slot $k$. We denote $\|\mathcal{P}\|_0$ as the number of nonzero entries in tensor $\mathcal{P}$. Since $\mathcal{P}$ is usually very sparse, we have $\|\mathcal{P}\|_0 \ll mnl$. Also, we assume that the $n$ items belong to $r$ item categories, with items in each category sharing similar inter-purchase durations.[3] We use an $n$-dimensional vector $\mathbf{c} \in \{1, 2, \ldots, r\}^n$ to represent the category membership of each item. Given $\mathcal{P}$ and $\mathbf{c}$, we further generate a tensor $\mathcal{T} \in \mathbb{R}^{m \times r \times l}$ where $t_{ic_jk}$ denotes the number of time slots between user $i$'s most recent purchase within item category $c_j$ until time $k$. If user $i$ has not purchased within item category $c_j$ until time $k$, $t_{ic_jk}$ is set to $+\infty$.

## 3.1 Inferring Purchase Intentions from Users' Purchase Histories

In this work, we formulate users' utility as a combined effect of form utility and time utility. To this end, we use an underlying third-order tensor $\mathcal{X} \in \mathbb{R}^{m \times n \times l}$ to quantify form utility. In addition, we employ a non-negative vector $\mathbf{d} \in \mathbb{R}_+^r$ to measure the underlying inter-purchase duration times of the $r$ item categories. It is understood that the inter-purchase durations for durable good categories are large, while for nondurable good categories are small, or even zero. In this study, we focus on items' inherent properties and assume that the inter-purchase durations are user-independent. The problem of learning personalized durations will be studied in our future work.

As discussed above, the demand is mediated by the time elapsed since the last purchase of an item in the same category. Let $d_{c_j}$ be the inter-purchase duration time of item $j$'s category $c_j$, and let $t_{ic_jk}$ be the time gap of user $i$'s most recent purchase within item category $c_j$ until time $k$. Then if $d_{c_j} > t_{ic_jk}$, a previously purchased item in category $c_j$ continues to be useful, and thus user $i$'s utility from item $j$ is weak. Intuitively, the greater the value $d_{c_j} - t_{ic_jk}$, the weaker the utility. On the other hand, $d_{c_j} < t_{ic_jk}$ indicates that the item is nearing the end of its lifetime and the user may be open to recommendations in category $c_j$. We use a hinge loss $\max(0, d_{c_j} - t_{ic_jk})$ to model such time utility. The overall utility can be obtained by comparing form utility and time utility. In more detail, we model a binary utility indicator tensor $\mathcal{Y} \in \{0, 1\}^{m \times n \times l}$ as being generated by the following thresholding process:

$$y_{ijk} = \mathbf{1}[x_{ijk} - \max(0, d_{c_j} - t_{ic_jk}) > \tau], \tag{1}$$

where $\mathbf{1}(\cdot) : \mathbb{R} \to \{0, 1\}$ is the indicator function, and $\tau > 0$ is a predefined threshold.

Note that the positive entries of $\mathcal{Y}$ denote high purchase intentions, while the positive entries of $\mathcal{P}$ denote actual purchases. Generally speaking, a purchase only happens when the utility is high, but a high utility does not necessarily lead to a purchase. This observation allows us to link the binary tensors $\mathcal{P}$ and $\mathcal{Y}$: $\mathcal{P}$ is generated by a one-sided sampling process that only reveals a subset of $\mathcal{Y}$'s positive entries. Given this observation, we follow [13] and include a label-dependent loss [26] trading the relative cost of positive and unlabeled samples:

$$\mathcal{L}(\mathcal{X}, \mathcal{P}) = \eta \sum_{ijk:\ p_{ijk}=1} \max[1 - (x_{ijk} - \max(0, d_{c_j} - t_{ic_jk})), 0]^2 + (1 - \eta) \sum_{ijk:\ p_{ijk}=0} l(x_{ijk}, 0),$$

where $l(x, c) = (x - c)^2$ denotes the squared loss.

In addition, the form utility tensor $\mathcal{X}$ should be of low-rank to capture temporal dynamics of users' interests, which are generally believed to be dictated by a small number of latent factors [22].

By combining asymmetric sampling and the low-rank property together, we jointly recover the tensor $\mathcal{X}$ and the inter-purchase duration vector $\mathbf{d}$ by solving the following tensor nuclear norm minimization (TNNM) problem:

$$\min_{\mathcal{X} \in \mathbb{R}^{m \times n \times l},\ \mathbf{d} \in \mathbb{R}_+^r} \eta \sum_{ijk:\ p_{ijk}=1} \max[1 - (x_{ijk} - \max(0, d_{c_j} - t_{ic_jk})), 0]^2$$
$$+ (1 - \eta) \sum_{ijk:\ p_{ijk}=0} x_{ijk}^2 + \lambda \|\mathcal{X}\|_*, \tag{2}$$

where $\|\mathcal{X}\|_*$ denotes the tensor nuclear norm, a convex combination of nuclear norms of $\mathcal{X}$'s unfolded matrices [21]. Given the learned $\hat{\mathcal{X}}$ and $\hat{\mathbf{d}}$, the underlying binary tensor $\mathcal{Y}$ can be recovered by (1).

We note that although the TNNM problem (2) can be solved by optimization techniques such as block coordinate descent [21] and ADMM [10], they suffer from high computational cost since they need to be solved iteratively with multiple SVDs at each iteration. An alternative way to solve the problem is tensor factorization [16]. However, this also involves iterative singular vector estimation and thus not scalable enough. As a typical example, recovering a rank 20 tensor of size $500 \times 500 \times 500$ takes the state-of-the-art tensor factorization algorithm TenALS [4] more than $20,000$ seconds on an Intel Xeon 2.40 GHz processor with 32 GB main memory.

## 3.2 A Scalable Relaxation

In this subsection, we discuss how to significantly improve the scalability of the proposed demand-aware recommendation model. To this end, we assume that an individual's form utility does not change over time, an assumption widely-used in many collaborative filtering methods [25, 32]. Under this assumption, the tensor $\mathcal{X}$ is a repeated copy of its frontal slice $\mathbf{x}_{::1}$, i.e.,

$$\mathcal{X} = \mathbf{x}_{::1} \circ \mathbf{e}, \tag{3}$$

where $\mathbf{e}$ is an $l$-dimensional all-one vector and the symbol $\circ$ represents the outer product operation. In this way, we can relax the problem of learning a third-order tensor $\mathcal{X}$ to the problem of learning its frontal slice, which is a second-order tensor (matrix). For notational simplicity, we use a matrix $\mathbf{X}$ to denote the frontal slice $\mathbf{x}_{::1}$, and use $x_{ij}$ to denote the entry $(i, j)$ of the matrix $\mathbf{X}$.

Since $\mathcal{X}$ is a low-rank tensor, its frontal slice $\mathbf{X}$ should be of low-rank as well. Hence, the minimization problem (2) simplifies to:

$$\min_{\substack{\mathbf{X} \in \mathbb{R}^{m \times n} \\ \mathbf{d} \in \mathbb{R}^r}} \eta \sum_{ijk:\ p_{ijk}=1} \max[1 - (x_{ij} - \max(0, d_{c_j} - t_{ic_jk})), 0]^2$$
$$+ (1 - \eta) \sum_{ijk:\ p_{ijk}=0} x_{ij}^2 + \lambda \|\mathbf{X}\|_* := f(\mathbf{X}, \mathbf{d}), \tag{4}$$

where $\|\mathbf{X}\|_*$ stands for the matrix nuclear norm, the convex surrogate of the matrix rank function. By relaxing the optimization problem (2) to the problem (4), we recover a matrix instead of a tensor to infer users' purchase intentions.

# 4   Optimization

Although the learning problem has been relaxed, optimizing (4) is still very challenging for two main reasons: (i) the objective is highly non-smooth with nested hinge losses, and (ii) it contains $mnl$ terms, and a naive optimization algorithm will take at least $O(mnl)$ time.

To address these challenges, we adopt an alternating minimization scheme that iteratively fixes one of $\mathbf{d}$ and $\mathbf{X}$ and minimizes with respect to the other. Specifically, we propose an extremely efficient optimization algorithm by effectively exploring the sparse structure of the tensor $\mathcal{P}$ and low-rank structure of the matrix $\mathbf{X}$. We show that (i) the problem (4) can be solved within $O(\|\mathcal{P}\|_0(k + \log(\|\mathcal{P}\|_0)) + (n + m)k^2)$ time, where $k$ is the rank of $\mathbf{X}$, and (ii) the algorithm converges to the critical points of $f(\mathbf{X}, \mathbf{d})$. In the following, we provide a sketch of the algorithm. The detailed description can be found in the supplementary material.

## 4.1   Update $\mathbf{d}$

When $\mathbf{X}$ is fixed, the optimization problem with respect to $\mathbf{d}$ can be written as:

$$\min_{\mathbf{d}} \sum_{ijk:\, p_{ijk}=1} \left\{ \max\left(1 - (x_{ij} - \max(0, d_{c_j} - t_{ic_jk})), 0\right)^2 \right\} := g(\mathbf{d}) := \sum_{ijk:\, p_{ijk}=1} g_{ijk}(d_{c_j}). \quad (5)$$

Problem (5) is non-trivial to solve since it involves nested hinge losses. Fortunately, by carefully analyzing the value of each term $g_{ijk}(d_{c_j})$, we can show that

$$g_{ijk}(d_{c_j}) = \begin{cases} \max(1 - x_{ij}, 0)^2, & \text{if } d_{c_j} \leq t_{ic_jk} + \max(x_{ij} - 1, 0) \\ (1 - (x_{ij} - d_{c_j} + t_{ic_jk}))^2, & \text{if } d_{c_j} > t_{ic_jk} + \max(x_{ij} - 1, 0). \end{cases}$$

For notational simplicity, we let $s_{ijk} = t_{ic_jk} + \max(x_{ij} - 1, 0)$ for all triplets $(i, j, k)$ satisfying $p_{ijk} = 1$. Now we can focus on each category $\kappa$: for each $\kappa$, we collect the set $Q = \{(i, j, k) \mid p_{ijk} = 1 \text{ and } c_j = \kappa\}$ and calculate the corresponding $s_{ijk}$s. We then sort $s_{ijk}$s such that $s_{(i_1j_1k_1)} \leq \cdots \leq s_{(i_{|Q|}j_{|Q|}k_{|Q|})}$. For each interval $[s_{(i_qj_qk_q)}, s_{(i_{q+1}j_{q+1}k_{q+1})}]$, the function is quadratic, thus can be solved in a closed form. Therefore, by scanning the solution regions from left to right according to the sorted $s$ values, and maintaining some intermediate computed variables, we are able to find the optimal solution, as summarized by the following lemma:

**Lemma 1.** *The subproblem* (5) *is convex with respect to* $\mathbf{d}$ *and can be solved exactly in* $O(\|\mathcal{P}\|_0 \log(\|\mathcal{P}\|_0))$, *where* $\|\mathcal{P}\|_0$ *is the number of nonzero elements in tensor* $\mathcal{P}$.

Therefore, we can efficiently update $\mathbf{d}$ since $\mathcal{P}$ is a very sparse tensor with only a small number of nonzero elements.

## 4.2   Update $\mathbf{X}$

By defining
$$a_{ijk} = \begin{cases} 1 + \max(0, d_{c_j} - t_{ic_jk}), & \text{if } p_{ijk} = 1 \\ 0, & \text{otherwise} \end{cases}$$

the subproblem with respect to $\mathbf{X}$ can be written as

$$\min_{\mathbf{X} \in \mathbb{R}^{m \times n}} h(\mathbf{X}) + \lambda\|\mathbf{X}\|_* \text{ where } h(\mathbf{X}) := \left\{ \eta \sum_{ijk:\, p_{ijk}=1} \max(a_{ijk} - x_{ij}, 0)^2 + (1 - \eta) \sum_{ijk:\, p_{ijk}=0} x_{ij}^2 \right\}.$$
$$(6)$$

Since there are $O(mnl)$ terms in the objective function, a naive implementation will take at least $O(mnl)$ time, which is computationally infeasible when the data is large. To address this issue, We use proximal gradient descent to solve the problem. At each iteration, $\mathbf{X}$ is updated by

$$\mathbf{X} \leftarrow S_\lambda(\mathbf{X} - \alpha \nabla h(\mathbf{X})), \quad (7)$$

where $S_\lambda(\cdot)$ is the soft-thresholding operator for singular values.[5]

Table 1: CPU time for solving problem (4) with different number of purchase records

| $m$ (# users) | $n$ (# items) | $l$ (# time slots) | $\|\boldsymbol{\mathcal{P}}\|_0$ | $k$ | CPU Time (in seconds) |
|---|---|---|---|---|---|
| 1,000,000 | 1,000,000 | 1,000 | 11,112,400 | 10 | 595 |
| 1,000,000 | 1,000,000 | 1,000 | 43,106,100 | 10 | 1,791 |
| 1,000,000 | 1,000,000 | 1,000 | 166,478,000 | 10 | 6,496 |

In order to efficiently compute the top singular vectors of $\mathbf{X} - \alpha \nabla h(\mathbf{X})$, we rewrite it as

$$\mathbf{X} - \alpha \nabla h(\mathbf{X}) = [1 - 2(1 - \eta)l]\,\mathbf{X} + \left(2(1-\eta) \sum_{ijk:\; p_{ijk}=1} x_{ij} - 2\eta \sum_{ijk:\; p_{ijk}=1} \max(a_{ijk} - x_{ij}, 0)\right).$$

$$= f_a(\mathbf{X}) + f_b(\mathbf{X}).$$

Since $f_a(\mathbf{X})$ is of low-rank and $f_b(\mathbf{X})$ is sparse, multiplying $(\mathbf{X} - \alpha \nabla h(\mathbf{X}))$ with a skinny $m$ by $k$ matrix can be computed in $O(nk^2 + mk^2 + \|\boldsymbol{\mathcal{P}}\|_0 k)$ time. As shown in [12], each iteration of proximal gradient descent for nuclear norm minimization only requires a fixed number of iterations of randomized SVD (or equivalently, power iterations) using the warm start strategy, thus we have the following lemma.

**Lemma 2.** *A proximal gradient descent algorithm can be applied to solve problem* (6) *within* $O(nk^2 T + mk^2 T + \|\boldsymbol{\mathcal{P}}\|_0 kT)$ *time, where $T$ is the number of iterations.*

We note that the algorithm is guaranteed to converge to the true solution. This is because when we apply a fixed number of iterations to update $\mathbf{X}$ via problem (7), it is equivalent to the "inexact gradient descent update" where each gradient is approximately computed and the approximation error is upper bounded by a constant between zero and one. Intuitively speaking, when the gradient converges to 0, the error will also converge to 0 at an even faster rate. See [12] for the detailed explanations.

### 4.3 Overall Algorithm

Combining the two subproblems together, the time complexity of each iteration of the proposed algorithm is:

$$O(\|\boldsymbol{\mathcal{P}}\|_0 \log(\|\boldsymbol{\mathcal{P}}\|_0) + nk^2 T + mk^2 T + \|\boldsymbol{\mathcal{P}}\|_0 kT).$$

**Remark:** Since each user should make at least one purchase and each item should be purchased at least once to be included in $\boldsymbol{\mathcal{P}}$, $n$ and $m$ are smaller than $\|\boldsymbol{\mathcal{P}}\|_0$. Also, since $k$ and $T$ are usually very small, the time complexity to solve problem (4) is dominated by the term $\|\boldsymbol{\mathcal{P}}\|_0$, which is a significant improvement over the naive approach with at least $O(mnl)$ complexity.

Since our problem has only two blocks $\mathbf{d}$, $\mathbf{X}$ and each subproblem is convex, our optimization algorithm is guaranteed to converge to a stationary point [11]. Indeed, it converges very fast in practice. As a concrete example, our experiment shows that it takes only 9 iterations to optimize a problem with 1 million users, 1 million items, and more than 166 million purchase records.

## 5 Experiments

### 5.1 Experiment with Synthesized Data

We first conduct experiments with simulated data to verify that the proposed demand-aware recommendation algorithm is computationally efficient and robust to noise. To this end, we first construct a low-rank matrix $\mathbf{X} = \mathbf{W}\mathbf{H}^T$, where $\mathbf{W} \in \mathbb{R}^{m \times 10}$ and $\mathbf{H} \in \mathbb{R}^{n \times 10}$ are random Gaussian matrices with entries drawn from $\mathcal{N}(1, 0.5)$, and then normalize $\mathbf{X}$ to the range of $[0, 1]$. We randomly assign all the $n$ items to $r$ categories, with their inter-purchase durations $\mathbf{d}$ equaling $[10, 20, \ldots, 10r]$. We then construct the high purchase intension set $\Omega = \{(i, j, k) \mid t_{ic_j k} \geq d_{c_j} \text{ and } x_{ij} \geq 0.5\}$, and sample a subset of its entries as the observed purchase records. We let $n = m$ and vary them in the range $\{10,000, 20,000, 30,000, 40,000\}$. We also vary $r$ in the range $\{10, 20, \cdots, 100\}$. Given the learned durations $\mathbf{d}^*$, we use $\|\mathbf{d} - \mathbf{d}^*\|_2 / \|\mathbf{d}\|_2$ to measure the prediction errors.

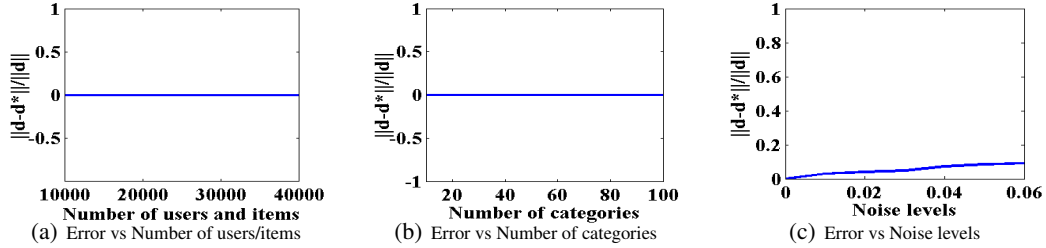

<div style="text-align:center">(a) Error vs Number of users/items     (b) Error vs Number of categories     (c) Error vs Noise levels</div>

Figure 1: Prediction errors $\|\mathbf{d} - \mathbf{d}^*\|_2/\|\mathbf{d}\|_2$ as a function of number of users, items, categories, and noise levels on synthetic datasets

**Accuracy** Figure 1(a) and 1(b) clearly show that the proposed algorithm can *perfectly* recover the underlying inter-purchase durations with varied numbers of users, items, and categories. To further evaluate the robustness of the proposed algorithm, we randomly flip some entries in tensor $\mathcal{P}$ from 0 to 1 to simulate the rare cases of purchasing two items in the same category in close temporal succession. Figure 1(c) shows that when the ratios of noisy entries are not large, the predicted durations $\hat{\mathbf{d}}$ are close enough to the true durations, thus verifying the robustness of the proposed algorithm.

**Scalability** To verify the scalability of the proposed algorithm, we fix the numbers of users and items to be 1 million, the number of time slots to be $1,000$, and vary the number of purchase records (i.e., $\|\mathcal{P}\|_0$). Table 1 summarizes the CPU time of solving problem (4) on an Intel Xeon 2.40 GHz server with 32 GB main memory. We observe that the proposed algorithm is extremely efficient, e.g., even with 1 million users, 1 million items, and more than 166 million purchase records, the running time of the proposed algorithm is less than 2 hours.

## 5.2 Experiment with Real-World Data

In the real-world experiments, we evaluate the proposed demand-aware recommendation algorithm by comparing it with the six state-of the-art recommendation methods: (a) $\mathbf{M^3F}$, maximum-margin matrix factorization [24], (b) **PMF**, probabilistic matrix factorization [25], (c) **WR-MF**, weighted regularized matrix factorization [14], (d) **CP-APR**, Candecomp-Parafac alternating Poisson regression [7], (e) **Rubik**, knowledge-guided tensor factorization and completion method [30], and (f) **BPTF**, Bayesian probabilistic tensor factorization [31]. Among them, $M^3F$ and PMF are widely-used static collaborative filtering algorithms. We include these two algorithms as baselines to justify whether traditional collaborative filtering algorithms are suitable for general e-commerce recommendation involving both durable and nondurable goods. Since they require explicit ratings as inputs, we follow [2] to generate numerical ratings based on the frequencies of (user, item) consumption pairs. WR-MF is essentially the positive-unlabeled version of PMF and has shown to be very effective in modeling implicit feedback data. All the other three baselines, i.e., CP-APR, Rubik, and BPTF, are tensor-based methods that can consider time utility when making recommendations. We refer to the proposed recommendation algorithm as **Demand-Aware Recommender for One-Sided Sampling**, or **DAROSS** for short.

Our testbeds are two real-world datasets *Tmall*[6] and *Amazon Review*[7]. Since some of the baseline algorithms are not scalable enough, we first conduct experiments on their subsets and then on the full set of *Amazon Review*. In order to generate the subsets, we randomly sample 80 item categories for *Tmall dataset* and select the users who have purchased at least 3 items within these categories, leading to the purchase records of 377 users and 572 items. For *Amazon Review dataset*, we randomly select 300 users who have provided reviews to at least 5 item categories on Amazon.com. This leads to a total of $5,111$ items belonging to 11 categories. Time information for both datasets is provided in days, and we have 177 and 749 time slots for *Tmall* and *Amazon Review* subsets, respectively. The full *Amazon Review* dataset is significantly larger than its subset. After removing duplicate items, it contains more than 72 million product reviews from 19.8 million users and 7.7 million items that

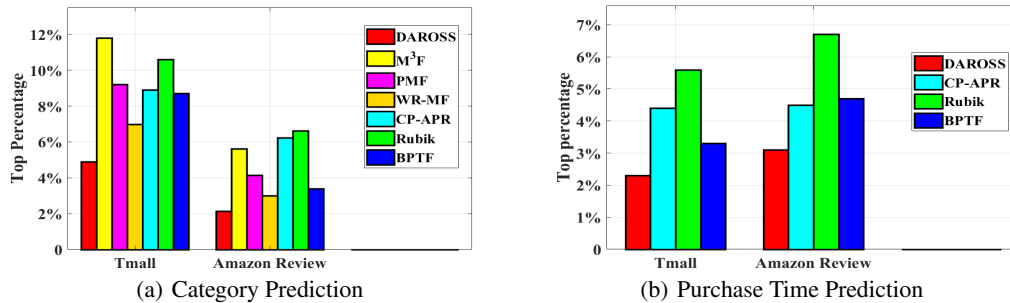

(a) Category Prediction　　　　　　　　(b) Purchase Time Prediction

Figure 2: Prediction performance on real-world datasets *Tmall* and *Amazon Review* subsets

Table 2: Estimated inter-review durations for Amazon Review subset

| Categories | Instant Video | Apps for Android | Automotive | Baby | Beauty | Digital Music | Grocery ... Food | Musical Instruments | Office Products | Patio ... Garden | Pet Supplies |
|---|---|---|---|---|---|---|---|---|---|---|---|
| **d** | 0 | 0 | 326 | 0 | 0 | 158 | 0 | 38 | 94 | 271 | 40 |

belong to 24 item categories. The collected reviews span a long range of time: from May 1996 to July 2014, which leads to $6,639$ time slots in total. Comparing to its subset, the full set is a much more challenging dataset both due to its much larger size and much higher sparsity, i.e., many reviewers only provided a few reviews, and many items were only reviewed a small number of times.

For each user, we randomly sample $90\%$ of her purchase records as the training data, and use the remaining $10\%$ as the test data. For each purchase record $(u, i, t)$ in the test set, we evaluate all the algorithms on two tasks: (i) *category prediction*, and (ii) *purchase time prediction*. In the first task, we record the highest ranking of items that are within item $i$'s category among all items at time $t$. Since a purchase record $(u, i, t)$ may suggest that in time slot $t$, user $u$ needed an item that share similar functionalities with item $i$, *category prediction* essentially checks whether the recommendation algorithms recognize this need. In the second task, we record the number of slots between the true purchase time $t$ and its nearest predicted purchase time within item $i$'s category. Ideally, good recommendations should have both small category rankings and small time errors. Thus we adopt the average top percentages, i.e., (*average category ranking) / $n$ $\times 100\%$* and (*average time error) / $l$ $\times 100\%$*, as the evaluation metrics of category and purchase time prediction tasks, respectively. The algorithms M³F, PMF, and WR-MF are excluded from the purchase time prediction task since they are static models that do not consider time information.

Figure 2 displays the predictive performance of the seven recommendation algorithms on *Tmall* and *Amazon Review* subsets. As expected, M³F and PMF fail to deliver strong performance since they neither take into account users' demands, nor consider the positive-unlabeled nature of the data. This is verified by the performance of WR-MF: it significantly outperforms M³F and PMF by considering the PU issue and obtains the second-best item prediction accuracy on both datasets (while being unable to provide a purchase time prediction). By taking into account both issues, our proposed algorithm DAROSS yields the best performance for both datasets and both tasks. Table 2 reports the *inter-review* durations of *Amazon Review* subset estimated by our algorithm. Although they may not perfectly reflect the true inter-purchase durations, the estimated durations clearly distinguish between durable good categories, e.g., *automotive*, *musical instruments*, and non-durable good categories, e.g., *instant video*, *apps*, and *food*. Indeed, the learned inter-purchase durations can also play an important role in applications more advanced than recommender systems, such as inventory management, operations management, and sales/marketing mechanisms. We do not report the estimated durations of *Tmall* herein since the item categories are anonymized in the dataset.

Finally, we conduct experiments on the full *Amazon Review* dataset. In this study, we replace *category prediction* with a more strict evaluation metric *item prediction* [8], which indicates the predicted ranking of item $i$ among all items at time $t$ for each purchase record $(u, i, t)$ in the test set. Since most of our baseline algorithms fail to handle such a large dataset, we only obtain the predictive performance of three algorithms: DAROSS, WR-MF, and PMF. Note that for such a large dataset, prediction time instead of training time becomes the bottleneck: to evaluate average item rankings, we

need to compute the scores of all the 7.7 million items, thus is computationally inefficient. Therefore, we only sample a subset of items for each user and estimate the rankings of her purchased items. Using this evaluation method, the average item ranking percentages for DAROSS, WR-MF, and PMF are $16.7\%$, $27.3\%$, and $38.4\%$, respectively. In addition to superior performance, it only takes our algorithm 10 iterations and 1 hour to converge to a good solution. Since WR-MF and PMF are both static models, our algorithm is the only approach evaluated here that considers time utility while being scalable enough to handle the full *Amazon Review* dataset. Note that this dataset has more users, items, and time slots but fewer purchase records than our largest synthesized dataset, and the running time of the former dataset is lower than the latter one. This clearly verifies that the time complexity of our algorithm is dominated by the number of purchase records instead of the tensor size. Interestingly, we found that some inter-review durations estimated from the full *Amazon Review* dataset are much smaller than the durations estimated from its subset. This is because the durations may be underestimated when many users reviewed items within a same durable goods category in close temporal succession. On the other hand, this result verifies the effectiveness of the PU formulation – even if the durations are underestimated, our algorithm still outperforms the competitors by a considerable margin. As a final note, we want to point out that *Tmall* and *Amazon Review* may not take full advantage of the proposed algorithm, since (i) their categories are relatively coarse and may contain multiple sub-categories with different durations, and (ii) the time stamps of *Amazon Review* reflect the review time instead of purchase time, and inter-review durations could be different from inter-purchase durations. By choosing a purchase history dataset with a more appropriate category granularity, we may obtain more accurate duration estimations and also a better recommendation performance.

## 6 Conclusion

In this paper, we examine the problem of demand-aware recommendation in settings when inter-purchase duration within item categories affects users' purchase intention in combination with intrinsic properties of the items themselves. We formulate it as a tensor nuclear norm minimization problem that seeks to jointly learn the form utility tensor and a vector of inter-purchase durations, and propose a scalable optimization algorithm with a tractable time complexity. Our empirical studies show that the proposed approach can yield perfect recovery of duration vectors in noiseless settings; it is robust to noise and scalable as analyzed theoretically. On two real-world datasets, *Tmall* and *Amazon Review*, we show that our algorithm outperforms six state-of-the-art recommendation algorithms on the tasks of category, item, and purchase time predictions.

## Acknowledgements

Cho-Jui Hsieh and Yao Li acknowledge the support of NSF IIS-1719097, TACC and Nvidia.

## Footnotes

*Now at Tencent AI Lab, Bellevue, WA, USA

[2]A detailed illustration can be found in the supplementary material

[3]To meet this requirement, the granularity of categories should be properly selected. For instance, the category "Smart TV" is a better choice than the category "Electrical Equipment", since the latter category covers a broad range of goods with different durations.

[4] http://web.engr.illinois.edu/~swoh/software/optspace/code.html

[5]If $\mathbf{X}$ has the singular value decomposition $\mathbf{X} = \mathbf{U}\Sigma\mathbf{V}^T$, then $\mathcal{S}_\lambda(\mathbf{X}) = \mathbf{U}(\Sigma - \lambda I)_+\mathbf{V}^T$ where $a_+ = \max(0, a)$.

[6]http://ijcai-15.org/index.php/repeat-buyers-prediction-competition

[7]http://jmcauley.ucsd.edu/data/amazon/

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
