[Supplementary Material]

# Supplementary Material:
# Scalable Demand-Aware Recommendation

Jinfeng Yi[1,*], Cho-Jui Hsieh[2], Kush R. Varshney[1], Lijun Zhang[3], Yao Li[2]

[1]IBM Thomas J. Watson Research Center, Yorktown Heights, NY, USA
[2]University of California, Davis, CA, USA
[3]National Key Laboratory for Novel Software Technology, Nanjing University, Nanjing, China
jinfengyi.ustc@gmail.com, chohsieh@ucdavis.edu, krvarshn@us.ibm.com,
zhanglj@lamda.nju.edu.cn, yaoli@ucdavis.edu

## 1   Illustration of Time Utility's Impact on Rank

To illustrate the point that the purchase intention matrix can be of high-rank, we construct a toy example with 50 users and 100 durable goods. As discussed in the main paper, user $i$'s purchase intention of item $j$ is mediated by a time utility factor $h_{ij}$, which is a function of item $j$'s inter-purchase duration $d$ and the time gap $t$ of user $i$'s most recent purchase within the item $j$'s category. If $d$ and $t$ are Gaussian random variables, then the time utility $h_{ij} = \max(0, d - t)$ follows a rectified Gaussian distribution. Following the widely adopted low-rank assumption, we also assume that the form utility matrix $\mathbf{X} \in \mathbb{R}^{50 \times 100}$ is generated by $\mathbf{U}\mathbf{V}^{\top}$, where $\mathbf{U} \in \mathbb{R}^{50 \times 10}$ and $\mathbf{V} \in \mathbb{R}^{100 \times 10}$ are both Gaussian random matrices. Here we assume that $\mathbf{U}$, $\mathbf{V}$, and the time utility matrix $\mathbf{H}$ share the same mean (= 1) and standard deviation (= 0.5). Given the form utility $\mathbf{X}$ and time utility $\mathbf{H}$, the purchase intention matrix $\mathbf{B} \in \mathbb{R}^{50 \times 100}$ is given by $\mathbf{B} = \mathbf{X} - \mathbf{H}$. Figure 1 shows the distributions of singular values for matrices $\mathbf{X}$ and $\mathbf{B}$. It clearly shows that although the form utility matrix $\mathbf{X}$ is of low-rank, the purchase intention matrix $\mathbf{B}$ is a full-rank matrix since all its singular values are greater than 0. This simple example illustrates that considering users' demands can make the underlying matrix no longer of low-rank, thus violating the key assumption made by many collaborative filtering algorithms.

Figure 1: A toy example that illustrates the impact of time utility. It shows that although the form utility matrix is of low-rank (rank 10), the purchase intention matrix is of full-rank (rank 50).

## 2 The Proposed Optimization Algorithm

In this section, we introduce how to efficiently optimize the following optimization problem:

$$\min_{\substack{\mathbf{X}\in\mathbb{R}^{m\times n} \\ \mathbf{d}\in\mathbb{R}^r}} \quad \eta \sum_{ijk:\; p_{ijk}=1} \max[1 - (x_{ij} - \max(0, d_{c_j} - t_{ic_jk})), 0]^2$$

$$+ (1 - \eta) \sum_{ijk:\; p_{ijk}=0} x_{ij}^2 + \lambda \left\| \mathbf{X} \right\|_* := f(\mathbf{X}, \mathbf{d}), \tag{1}$$

We note that optimizing (1) is a very challenging problem for two reasons: (i) the objective is highly non-smooth with nested hinge losses, and (ii) it contains $mnl$ terms: a naive optimization algorithm will take at least $O(mnl)$ time.

To address these challenges, we adopt an alternating minimization scheme that iteratively fixes one of $\mathbf{d}$ and $\mathbf{X}$ and minimizes with respect to the other. Specifically, we apply an alternating minimization scheme to iteratively solve the following subproblems:

$$\mathbf{d} \leftarrow \arg\min_{\mathbf{d}} f(\mathbf{X}, \mathbf{d}). \tag{2}$$

$$\mathbf{X} \leftarrow \arg\min_{\mathbf{X}} f(\mathbf{X}, \mathbf{d}) \tag{3}$$

We note that both subproblems are non-trivial to solve because subproblem (3) is a nuclear norm minimization problem, and both subproblems involve nested hinge losses. In the following, we discuss how to efficiently optimize subproblems (2) and (3):

### 2.1 Update d

Eq (2) can be written as

$$\min_{\mathbf{d}} \sum_{ijk:\; p_{ijk}=1} \left\{ \max\left(1 - (x_{ij} - \max(0, d_{c_j} - t_{ic_jk})), 0\right)^2 \right\} := g(\mathbf{d}) := \sum_{ijk:\; p_{ijk}=1} g_{ijk}(d_{c_j}).$$

We then analyze the value of each $g_{ijk}$ by comparing $d_{c_j}$ and $t_{ic_jk}$:

1. If $d_{c_j} \leq t_{ic_jk}$, we have
$$g_{ijk}(d_{c_j}) = \max(1 - x_{ij}, 0)^2$$

2. If $d_{c_j} > t_{ic_jk}$, we have
$$g_{ijk}(d_{c_j}) = \max(1 - (x_{ij} - d_{c_j} + t_{ic_jk}), 0)^2,$$

   which can be further separated into two cases:
$$g_{ijk}(d_{c_j}) = \begin{cases} 1 - (x_{ij} - d_{c_j} + t_{ic_jk}))^2, & \text{if } d_{c_j} > x_{ij} + t_{ic_jk} - 1 \\ 0, & \text{if } d_{c_j} \leq x_{ij} + t_{ic_jk} - 1 \end{cases}$$

Therefore, we have the following observations:

1. If $x_{ij} \leq 1$, we have
$$g_{ijk}(d_{c_j}) = \begin{cases} \max(1 - x_{i,j}, 0)^2, & \text{if } d_{c_j} \leq t_{ic_jk} \\ (1 - (x_{ij} - d_{c_j} + t_{ic_jk}))^2, & \text{if } d_{c_j} > t_{ic_jk} \end{cases}$$

2. If $x_{ij} > 1$, we have
$$g_{ijk}(d_{c_j}) = \begin{cases} (1 - (x_{ij} - d_{c_j} + t_{ic_jk}))^2, & \text{if } d_{c_j} > t_{ic_jk} + x_{ij} - 1 \\ 0, & \text{if } d_{c_j} \leq t_{ic_jk} + x_{ij} - 1 \end{cases}$$

This further implies

$$g_{ijk}(d_{c_j}) = \begin{cases} \max(1 - x_{ij}, 0)^2, & \text{if } d_{c_j} \leq t_{ic_jk} + \max(x_{ij} - 1, 0) \\ (1 - (x_{ij} - d_{c_j} + t_{ic_jk}))^2, & \text{if } d_{c_j} > t_{ic_jk} + \max(x_{ij} - 1, 0) \end{cases}$$

For notational simplicity, we let $s_{ijk} = t_{ic_jk} + \max(x_{ij} - 1, 0)$ for all triplets $(i, j, k)$ satisfying $p_{ijk} = 1$.

**Algorithm.** For each category $\kappa$, we collect the set $Q = \{(i, j, k) \mid p_{ijk} = 1 \text{ and } c_j = \kappa\}$ and calculate the corresponding $s_{ijk}$s. We then sort $s_{ijk}$s such that $s_{(i_1j_1k_1)} \leq \cdots \leq s_{(i_{|Q|}j_{|Q|}k_{|Q|})}$. For each interval $[s_{(i_qj_qk_q)}, s_{(i_{q+1}j_{q+1}k_{q+1})}]$, the function is

$$g_\kappa(d) = \sum_{t=q+1}^{|Q|} \max(1 - x_{i_tj_t}, 0)^2 + \sum_{t=1}^{q}(d + 1 - x_{i_tj_t} - t_{i_tc_{j_t}k_t})^2$$

By letting

$$R_q = \sum_{t=q+1}^{|Q|} \max(1 - x_{i_tj_t}, 0)^2,$$

$$F_q = \sum_{t=1}^{q}(1 - x_{i_tj_t} - t_{i_tc_{j_t}k_t}),$$

$$W_q = \sum_{t=1}^{q}(1 - x_{i_tj_t} - t_{i_tc_{j_t}k_t})^2,$$

we have

$$g_\kappa(d) = qd^2 + 2F_qd + W_q + R_q$$
$$= q\left(d + \frac{F_q}{q}\right)^2 - \frac{F_q^2}{q} + W_q + R_q.$$

Thus the optimal solution in the interval $[s_{(i_qj_qk_q)}, s_{(i_{q+1}j_{q+1}k_{q+1})}]$ is given by

$$d^* = \max\left(s_{(i_qj_qk_q)}, \min\left(s_{(i_{q+1}j_{q+1}k_{q+1})}, -\frac{F_q}{q}\right)\right),$$

and the optimal function value is $g_r(d^*)$. By going through all the intervals from small to large, we can obtain the optimal solution for the whole function. We note that each time when $q \Rightarrow q + 1$, the constants $R_q, F_q, W_q$ only change by one element. Thus the time complexity for going from $q \Rightarrow q + 1$ is $O(1)$, and the whole procedure has a time complexity $O(|Q|)$.

In summary, we can solve the subproblem (2) by the following steps:

1. generate the set $Q_\kappa = \{(i, j, k) \mid p_{ijk} = 1 \text{ and } c_j = \kappa\}$ for each category $r$,
2. sort each list (costing $O(|Q_\kappa| \log |Q_\kappa|)$ time),
3. compute $R_0, F_0, W_0$ (costing $O(|Q_\kappa|)$ time), and then
4. search for the optimal solution for each $q = 1, 2, \cdots, |Q_\kappa|$ (costing $O(|Q_\kappa|)$ time).

The above steps lead to an overall time complexity $O(\|\mathcal{P}\|_0 \log(\|\mathcal{P}\|_0))$, where $\|\mathcal{P}\|_0$ is the number of nonzero elements in tensor $\mathcal{P}$. Therefore, we can efficiently update $\mathbf{d}$ since $\mathcal{P}$ is a very sparse tensor with only a small number of nonzero elements.

## 2.2 Update X

By defining

$$a_{ijk} = \begin{cases} 1 + \max(0, d_{c_j} - t_{ic_jk}), & \text{if } p_{ijk} = 1 \\ 0, & \text{otherwise} \end{cases}$$

---
**Algorithm 1:** Proximal Gradient Descent for Updating $\mathbf{X}$
---
**Input** : $\mathcal{P}$, $\mathbf{X}^0$ (initialization), step size $\gamma$
**Output** : A sequence of $\mathbf{X}^t$ converges to the optimal solution

**1 for** $t = 1, \ldots, maxiter$ **do**
**2**    $[\mathbf{U}, \Sigma, \mathbf{V}] = \text{rand\_svd}(\mathbf{X} - \gamma \nabla h(\mathbf{X}^t))$
**3**    $\bar{\Sigma} = \max(\Sigma - \gamma\lambda, 0)$
**4**    $k$ : number of nonzeros in $\Sigma$
**5**    $\mathbf{X}^{t+1} = \mathbf{U}(:, 1{:}k)\bar{\Sigma}(1{:}k, 1{:}k)\mathbf{V}(:, 1{:}k)^T$
---

the subproblem (3) can be written as

$$\min_{\mathbf{X} \in \mathbb{R}^{m \times n}} h(\mathbf{X}) + \lambda\|\mathbf{X}\|_* \text{ where } h(\mathbf{X}) := \left\{ \eta \sum_{ijk:\ p_{ijk}=1} \max(a_{ijk} - x_{ij}, 0)^2 + (1 - \eta) \sum_{ijk:\ p_{ijk}=0} x_{ij}^2 \right\}.$$

Since there are $O(mnl)$ terms in the objective function, a naive implementation will take $O(mnl)$ time, which is computationally infeasible when the data is large. To address this issue, We use proximal gradient descent to solve the problem. At each iteration, $\mathbf{X}$ is updated by

$$\mathbf{X} \leftarrow S_\lambda(\mathbf{X} - \alpha \nabla h(\mathbf{X})), \tag{4}$$

where $S_\lambda(\cdot)$ is the soft-thresholding operator for singular values [2].

In order to efficiently compute the top singular vectors of $\mathbf{X} - \alpha \nabla h(\mathbf{X})$, we rewrite it as

$$\mathbf{X} - \alpha \nabla h(\mathbf{X}) = [1 - 2(1-\eta)l]\,\mathbf{X} + \left( 2(1-\eta) \sum_{ijk:\ p_{ijk}=1} x_{ij} - 2\eta \sum_{ijk:\ p_{ijk}=1} \max(a_{ijk} - x_{ij}, 0) \right). \tag{5}$$

Since $\mathbf{X}$ is a low-rank matrix, $[1 - 2(1 - \eta)l]\,\mathbf{X}$ is also of low-rank. Besides, since $\mathcal{P}$ is very sparse, the term

$$\left( 2(1-\eta) \sum_{ijk:\ p_{ijk}=1} x_{ij} - 2\eta \sum_{ijk:\ p_{ijk}=1} \max(a_{ijk} - x_{ij}, 0) \right)$$

is also sparse because it only involves the nonzero elements of $\mathcal{P}$. In this case, when we multiply $(\mathbf{X} - \alpha \nabla h(\mathbf{X}))$ with a skinny $m$ by $k$ matrix, it can be computed in $O(nk^2 + mk^2 + \|\mathcal{P}\|_0 k)$ time.

As shown in [1], each iteration of proximal gradient descent for nuclear norm minimization only requires a fixed number of iterations before convergence, thus the time complexity to update $\mathbf{X}$ is $O(nk^2 T + mk^2 T + \|\mathcal{P}\|_0 kT)$, where $T$ is the number of iterations.

Since each user should make at least one purchase and each item should be purchased at least once to be included in $\mathcal{P}$, $n$ and $m$ are smaller than $\|\mathcal{P}\|_0$. Also, since $k$ and $T$ are usually very small, the time complexity to solve problem (3) is dominated by the term $\|\mathcal{P}\|_0$, which is a significant improvement over the naive approach with $O(mnl)$ complexity.

## Footnotes

*Now at Tencent AI Lab, Bellevue, WA, USA

[2]If $\mathbf{X}$ has the singular value decomposition $\mathbf{X} = \mathbf{U}\Sigma\mathbf{V}^T$, then $\mathcal{S}_\lambda(\mathbf{X}) = \mathbf{U}(\Sigma - \lambda I)_+ \mathbf{V}^T$ where $a_+ = \max(0, a)$.

## References

[1] C.-J. Hsieh and P. A. Olsen. Nuclear norm minimization via active subspace selection. In *ICML*, 2014.