[Reviews · NeurIPS 2017]

Reviewer 1



This paper proposed an scalable and accurate demand-aware recommendation system which considers the form utility and time utility, i.e., recommending related item at the correct time to the user. The paper has written very well and technically sounds correct. The experimental section is complete as it compares to several approaches and applied it to a large scale problem. The optimization is explained very well and utilized the structure of the matrix X to boost the performance of the algorithm (linear in terms of number of non zero).

Reviewer 2



Paper revolves over the observation that in e-commerce world customers rarely purchase two items that belong to the same category (e.g. Smartphone) in close temporal succession. Therefore, they claim that a robust recommendation system should incorporate both utility and time utility. An additional problem that is tackled in the paper is that many e-commerce systems have no explicit negative feedback to learn from (for example one can see only what items customer purchased - positives - and no explicit negatives in form of items user did not like). I believe that the second problem they mention is not as big of a concern as advertised by authors. In absence of any explicit negative signal good replacements are long dwell-time clicks that did not end up in purchase, as well as cart additions that did not end up in the final purchase or returns. Many companies are also implementing swipe to dismiss that is useful for collecting explicit negative signal and can be applied to any e-commerce site easily. The authors clearly define and describe the objective of modeling user intent as combined effect of form utility and time utility and identify limitations of minimizing the objective when formulated in the simplest form (as tensor nuclear norm) in terms of high computational cost. To tackle that the authors propose to relax the problem to a matrix optimization problem with a label-dependent loss and propose an efficient alternating minimization algorithm whose complexity is proportional to the number of purchase records in the matrix. I see the biggest strength of this paper as reformulating the objective and adding relaxations that significantly reduce computational cost and still achieves good result. Introducing time component adds another dimension and as you mentioned it results in a significant rise in computational cost O(mnl) where l is the number of time slots. What happens when you drop the time component in terms of accuracy? I saw that in your experiment you compare to many traditional collaborative filtering algorithms but you do not compare to your algorithm that doesn't take time in account or uses different definitions of what time slot is (e.g. just a s a two level definition it could be: purchased this month or not). Related work is generally covered well. Experiments only cover offline evaluation. Therefore, we have no sense of how this algorithm would work in practice in a production system. For example I would be curious to compare against just showing popular items as recommendations (as done in this paper: E-commerce in your inbox: Product recommendations at scale, KDD 2015). Paper would be much stronger with such results. Also, additional evaluation metrics should be considered, like traditional ranking metrics, precision at K, etc. It would be also interesting to put this in context of ad retargeting. It is well known that users on the Web are re-targeted for the same product they bought (or same category) even after they bought the product. While some see this as a waste of ad dollars some claim this still brings in revenue as users often want to see items from same category in order to get reassurance that you bought the best product from the category. In practice, it also happens that they return the original product and purchase the recommended one from same category. In practice, recommender production systems take this signal in account as well.

Reviewer 3



This is a strong paper, clearly written, well structured, and technically sound with elaborate mathematical derivations and sufficient supporting evidence gathered. On a conceptual level the mathematics isn’t ground-breaking, but what it might lack in elegance it makes up for in efficiency and performance of the algorithm. That is to say, it’s an interesting and solid piece of work from an engineering perspective. Some more emphasis might be placed on the weaknesses. For instance, in the two sentences on line 273-277, there's a rather brief remark on why the inter-purchase durations aren't perfectly modeled by the learned 'd' parameters. I think this is a missed opportunity at best, if not a slight omission. In particular, I believe that one of the interesting aspects of this work would that we can explicitly inspect the 'd' parameters (if they were meaningful). According to the author, the only thing to learn from them is which item category is durable/non-durable. The main aspect that I believe is innovative is the performance of the algorithm on large datasets (as the title suggests). As I just mentioned, a by-product of this approach is it explicitly models the inter-purchase duration for different item categories, which may be useful by itself in a commercial setting. The approach taken in this paper is quite pragmatic (as opposed to conceptual). This means that it's not overwhelmingly exciting (or reusable) on a conceptual level. Nevertheless, it does seem to push the envelop in terms of scalability of modeling (time-aware) repeat consumption. In terms of clarity, this paper is written in a way that makes implementation relatively painless. Moreover, a link to the author's implementation is provided in one of the Suggestions In section 2.1, eq. (1), the predictions y_ijk depend on a threshold \tau. In the optimization problem, however, we set out to solve the non-thresholded version of the predictions. This is not necessarily wrong, but I believe it warrants some explanation. In section 3.1, I would suggest to make a choice between either removing lines 163-169 or (better) include more details of the computation. I got lost there and needed the appendix in order to proceed. In other words, either defer those details to the appendix or explain more fully. In section 3.2, I believe it would be nice to add a few words on why the proposed proximal X updates will converge to the original optimization problem. Of course, there is a citation [10], but it would be nice to be a little bit more self-contained. One or two sentences will do. Minor typos line 166: reposition some curly brackets in your latex for the 's' subscripts. appendix line 51: replace U by Q Some remarks Have you considered model time directly as one leg in a type-3 tensor similar to your x_ijk (or at the level of category x_ick) and thereby removing the "d - t" term? Here, the index k could be given not as time, but as time-since-last-purchase (perhaps binned and indexed). This way, you also model "d - t" for instance x_{ic:} would give you a something that might be interpreted as time-utility defined over inter-purchase durations, given a user i and item category c.